# Consistency in the Assessment of Dried Blood Spot Specimen Size and Quality in U.K. Newborn Screening Laboratories

**DOI:** 10.3390/ijns10030060

**Published:** 2024-09-05

**Authors:** Stuart J. Moat, James R. Bonham, Christine Cavanagh, Margaret Birch, Caroline Griffith, Lynette Shakespeare, Clare Le Masurier, Claire Manfredonia, Beverly Hird, Philippa Goddard, Sarah Smith, Laura Wainwright, Rachel S. Carling, Jennifer Cundick, Fiona Jenkinson, Catherine Collingwood, Nick Flynn, Nazia Taj, Mehdi Mirzazadeh, Tejswurree Ramgoolam, Liz Robinson, Amy Headley, Tessa Morgan, David Elliman, Lesley Tetlow

**Affiliations:** 1Wales Newborn Screening Laboratory, Department of Medical Biochemistry, Immunology & Toxicology, University Hospital of Wales, Cardiff CF14 4XW, UK; 2University Hospital Wales, School of Medicine, Cardiff University, Cardiff CF14 4XW, UK; 3Clinical Chemistry, Sheffield Children’s NHSFT, Sheffield S10 2TH, UK; 4NHS England, Wellington House, London SE1 8UG, UK; 5Health Protection & Screening Services, Public Health Wales, Cardiff CF10 4BZ, UK; 6Biochemical Genetics, Specialist Laboratory Medicine, St James University Hospital, Leeds LS9 7TF, UK; 7South West Newborn Screening and Metabolic Laboratory, Severn Pathology, Southmead Hospital, Bristol BS10 5NB, UK; 8Newborn Screening Laboratory, Manchester University NHSFT, Manchester M13 9WL, UK; 9Newborn Screening and Biochemical Genetics, Paediatric Laboratory Medicine, Birmingham Children’s Hospital, Steelhouse Lane, Birmingham B4 6NH, UK; 10Scottish Newborn Screening Laboratory, Queen Elizabeth University Hospital, Glasgow G51 4TF, UK; 11Blood Sciences, Portsmouth Hospitals Trust, Portsmouth PO6 3LY, UK; 12Biochemical Sciences, Synnovis, Guys & St Thomas’ NHSFT, London SE1 7EH, UK; rachel.carling@viapath.co.uk; 13GKT School of Medical Education, Kings College London, London WC2R 2LS, UK; 14Regional Newborn Screening Laboratory, Royal Victoria Hospital, Belfast BT12 6BA, UK; 15Newborn Screening, Blood Sciences, Royal Victoria Infirmary, Newcastle upon Tyne NE1 4LP, UK; 16Biochemistry Department, Alder Hey Children’s NHS Foundation Trust, Liverpool L12 2AP, UK; 17Biochemical Genetics Unit, Cambridge University Hospitals NHSFT, Cambridge CB2 0QQ, UK; 18Oxford Screening Laboratory, Clinical Biochemistry, Oxford University Hospitals NHSFT, Oxford OX3 9DU, UK; 19South West Thames Newborn Screening, Epsom & St Helier Hospitals, Carshalton SM5 1AA, UK; 20Department of Chemical Pathology, Great Ormond Street Hospital for Children, London WC1N 3JH, UK

**Keywords:** filter paper, specimen collection device, dried blood spots, specimen collection, blood spot quality, avoidable repeat rate, external quality assessment

## Abstract

In 2015, U.K. newborn screening (NBS) laboratory guidelines were introduced to standardize dried blood spot (DBS) specimen quality acceptance and specify a minimum acceptable DBS diameter of ≥7 mm. The UK ‘acceptable’ avoidable repeat rate (AVRR) is ≤2%. To assess inter-laboratory variability in specimen acceptance/rejection, two sets of colored scanned images (*n* = 40/set) of both good and poor-quality DBS specimens were distributed to all 16 U.K. NBS laboratories for evaluation as part of an external quality assurance (EQA) assessment. The mean (range) number of specimens rejected in the first EQA distribution was 7 (1–16) and in the second EQA distribution was 7 (0–16), demonstrating that adherence to the 2015 guidelines was highly variable. A new minimum standard for DBS size of ≥8 mm (to enable a minimum of six sub-punches from two DBS) was discussed. NBS laboratories undertook a prospective audit and demonstrated that using ≥8 mm as the minimum acceptable DBS diameter would increase the AVRR from 2.1% (range 0.55% to 5.5%) to 7.8% (range 0.55% to 22.7%). A significant inverse association between the number of specimens rejected in the DBS EQA distributions and the predicted AVVR (using ≥8 mm minimum standard) was observed (r = −0.734, *p* = 0.003). Before implementing more stringent standards, the impact of a standard operating procedure (SOP) designed to enable a standardized approach of visual assessment and using the existing ≥7 mm diameter (to enable a minimum of four sub-punches from two DBS) as the minimum standard was assessed in a retrospective audit. Implementation of the SOP and using the ≥7 mm DBS diameter would increase the AVRR from 2.3% (range 0.63% to 5.3%) to 6.5% (range 4.3% to 20.9%). The results demonstrate that there is inconsistency in applying the acceptance/rejection criteria, and that a low AVVR is not an indication of good-quality specimens being received into laboratories. Further work is underway to introduce and maintain standards without increasing the AVRR to unacceptable levels.

## 1. Introduction

The process of dried blood spot (DBS) specimen collection typically involves the application of a non-volumetric amount of blood (single-hanging drop of blood) from a heel prick that disperses by spreading radially across the filter paper whilst penetrating the porous fibers to fully soak through the filter paper. In the U.K. (England, Wales, Scotland & Northern Ireland), the filter paper collection devices used for newborn screening (NBS) are manufactured using Revvity 226 grade filter paper. The U.K DBS collection devices have four printed circles with an inner diameter of 10 mm to serve as a guide for specimen collectors to obtain appropriate-sized specimens. When appropriately filled, a 10 mm circle contains approximately 35–50 μL of whole blood [1,2]. A good-quality specimen is one that contains sufficient blood, which has been applied correctly to the filter paper, to allow testing for all conditions (including confirmatory and second tier tests) to ensure that accurate screening results are obtained. In the U.K., there are 16 NBS laboratories (13 in England, 1 in Wales, 1 in Scotland and 1 in Northern Ireland) with ~700,000 babies screened per year for nine conditions. The majority of DBS specimens are collected by community midwives visiting homes between day 4 to 8 of life (birth = day 0).

The size and quality of DBS specimens received into NBS laboratories are assessed subjectively by visual inspection, ensuring that the printed circle is suitably filled with blood, that the blood is spread symmetrically and evenly with blood when viewed from both sides of the filter paper. Repeat specimens are requested on those specimens deemed unsuitable for analysis. Such specimens are termed “avoidable repeats”, and U.K. standards recommend that the ‘acceptable’ avoidable repeat rate (AVRR) should be ≤2% and that the ‘achievable’ AVVR should be <1%. In 2018/2019 (fiscal year), the mean AVRR in the U.K. was 2.7% (range 1.4–8.3%) [3], which resulted in ~20,000 requests for repeat specimens. Avoidable repeats can cause anxiety and harms for both infants and parents and delay the screening for life-threatening conditions, where turnaround time is a critical factor.

In a comprehensive evidence-based study investigating the impact of DBS variance factors [2], it was recommended that the following should be rejected: compressed specimens; specimens <20 µL (<8 mm diameter); insufficient and multi-spotted specimens. In 2014, the U.K. Newborn Screening Laboratory Network (UKNSLN) advised that there was a need to standardize DBS specimen acceptance and rejection criteria and that a U.K.-wide approach was required. Whilst it was recommended that an 8 mm DBS diameter should be used as the minimum acceptance criteria, it was deemed by some NBS Laboratory Directors that this would be too challenging. A compromise was made to accept a minimum DBS diameter of 7 mm from which two sub-punches of 3.2 mm in diameter could be taken. All U.K NBS laboratories use the Revvity Panthera puncher.

In 2015, an agreed set of guidelines was published [Appendix A], and following a period of focused training and assessment for specimen collectors and laboratory staff, they were then introduced with the aim of improving the quality of DBS specimens across the U.K., achieving an AVRR < 2% and standardizing the DBS specimen quality acceptance/rejection criteria to ensure the harmonization of DBS specimen rejection across the U.K. As part of the U.K. performance data analysis, the AVRRs are used to compare DBS quality across all 16 laboratories. It is important to recognize that a low AVRR does not necessarily indicate that good-quality specimens are being collected and accepted for analysis. However, at the time, there was no mechanism by which to assess blood spot acceptance/rejection performance between laboratories.

To date, no study has been undertaken to assess the impact of these new guidelines on improving DBS specimen quality in the U.K. and whether or not a low AVRR is an accurate indicator of good-quality specimens being collected. The aim of this study was to develop a DBS external quality assessment (EQA) scheme to investigate whether there was consistency between UK laboratories in the acceptance/rejection of specimens.

## 2. Materials and Methods

### 2.1. Assessment of the Consistency in DBS Specimen Acceptance/Rejection in UK NBS Laboratories Using the 2015 Specimen Acceptability Criteria

To assess variability in specimen acceptance/rejection rates in the U.K. NBS laboratories, scanned images of both good and poor-quality specimens and DBS of varying sizes were produced. These color images (front and back) were collated from routine DBS specimens received into the Wales NBS laboratory and other specimens that were contrived using residual anonymized liquid blood to simulate poor quality specimens and DBS of various sizes.

A 200 dpi color image of the DBS specimens (containing all four circles on the filter paper collection device, both front and back) was obtained using a desk top scanner (Fujitsu fi-6140 scanner) linked to Microsoft Paint. Images were cropped and then exported into Microsoft PowerPoint. The images produced were the exact size as routine specimens (i.e., 10 mm printed circles) and could therefore be placed under a blood spot puncher to assess DBS diameter and sample acceptance.

Two sets of images were created, and each set contained 40 images of DBS specimens on the filter paper collection device. DBS EQA image set 1 was distributed in March 2019 and DBS EQA image set 2 in May 2019. Image sets were sent to all 16 U.K. NBS laboratories. Results from the two distributions were not provided until results had been received from all participating laboratories to ensure that laboratories did not alter current practice in specimen acceptance/rejection.

### 2.2. Establishment of the U.K. DBS Quality Group

In response to the findings from the initial EQA assessment study, a DBS quality group was convened to establish an up-to-date evidence-based minimum standard for DBS specimen size and quality to ensure that reliable results are produced and to describe a consistent way in which to introduce and maintain such standards without increasing the AVRR to unacceptable levels.

#### 2.2.1. Development of a Standard Operating Procedure (SOP) for DBS Specimen Acceptance/Rejection

Updated DBS specimen acceptability guidelines were developed based upon published peer-reviewed studies and the NBS01 Clinical and Laboratory Standards Institute standards for the collection of DBS specimens for newborn screening [1,2,4,5,6,7,8,9,10,11,12,13,14,15]. It was agreed that the minimum and maximum acceptable DBS diameter would be ≥8 and ≤14 mm, respectively. A minimum DBS diameter of ≥8 mm would enable a minimum of six sub-punches to be taken from two DBS. Standardized puncher settings (All U.K. NBS laboratories use the Revvity Panthera Puncher) were used in order to be able to select three 3.2 mm sub-punches as a proxy for an 8 mm DBS diameter to guide minimum DBS size rejection. An 8 mm DBS will provide three 3.2 mm discs for testing, and a minimum of two DBS ≥ 8 mm or one larger spot (≥8 mm but ≤14 mm) will enable initial analysis and allow re-testing in duplicate if required. In the U.K., a minimum of four sub-punches are required to undertake initial screening (1 for inherited metabolic disorders, 1 for cystic fibrosis, 1 for congenital hypothyroidism and 1 for sickle cell disorders).

In addition, a standard operating procedure (SOP) was also developed that contained images of good and poor-quality specimens to assist laboratories with identifying and appropriately classifying poor-quality specimens in order to enable a standardized approach of visual assessment of specimens. Throughout this process, input and guidance was sought from laboratory leads and midwifery colleagues.

#### 2.2.2. Assessment of the Impact of Moving to a Minimum DBS Diameter Size of ≥8 mm for Specimen Rejection on the AVRR in UK NBS Laboratories

To assess the potential impact of using ≥8 mm as the minimum DBS diameter, laboratories were asked to prospectively assess DBS specimens being routinely received during the first two weeks of November 2020. Laboratories were specifically asked to collate data on the number of routine specimens received, the number rejected using current laboratory practice and the number of specimens that would have been rejected if two ≥8 mm diameter DBS were used as the minimum standard for DBS acceptability.

#### 2.2.3. Assessment of the Impact of Utilizing the National SOP Using the Existing Minimum DBS Diameter of ≥7 mm to Guide Specimen Acceptance/Rejection in UK NBS Laboratories

Concerns were raised that the AVRR would increase to unacceptable levels in certain areas of the U.K. if the minimum DBS diameter was increased to ≥8 mm. The DBS Quality group was cautious about implementing more stringent rejection criteria until a consistent approach to introduce and maintain such standards without increasing the AVRR to unacceptable levels (such as education and training) had been outlined. It was agreed that instead of moving to a minimum acceptable diameter of ≥8 mm, it was decided to modify the SOP to include the existing minimum acceptable diameter of ≥7 mm and to then assess the impact of the SOP to improve the consistency with which the laboratories assess quality.

The new SOP which included numerous color images of poor-quality DBS specimens and using ≥7 mm as the minimum acceptable diameter (i.e., minimum of 4 sub-punches to be taken from two DBS) and >14 mm as the maximal diameter was sent out to all laboratories. The laboratories were requested to retrospectively assess the potential impact of the SOP on the AVRR during the month of November 2022. Please see Appendix A for the SOP for assessing DBS specimen acceptability.

## 3. Results

### 3.1. Assessment of the Consistency in DBS Specimen Acceptance/Rejection in U.K NBS Laboratories Using the 2015 Specimen Acceptability Criteria

The results of the initial two EQA distributions are shown in Figure 1a,b. In distribution 1, a total of 15 out of the 16 U.K. laboratories agreed to participate, and all 15 laboratories returned results. The mean (range) number of specimens rejected was 7 (1 to 16). Six laboratories rejected 5 or less specimens, and 4 rejected ≥10 specimens. There were 5 specimens with a clear consensus for rejection (i.e., >60%). In distribution 2, a total of 15 out of the 16 U.K. laboratories agreed to participate in this EQA exercise, and 14/15 laboratories returned results. The mean (range) number of specimens rejected was 7 (0 to 16). Five laboratories rejected 5 or less specimens, and 6 rejected ≥10 specimens. There were 5 specimens with a clear consensus for rejection (i.e., >60%). Examples of some of the images distributed as part of the EQA scheme are shown in Figure 2.

No association was found between the mean number of specimens rejected in the DBS quality EQA distributions 1 and 2 and the UK AVVR (%) in the U.K. during the period of 2018–2019; r = 0.194, *p* = 0.506 (Figure 3).

### 3.2. Assessment of the Impact of Using ≥8 mm as the Minimum Acceptable DBS Diameter on the AVRR in the U.K. NBS Laboratories

The impact of moving from a minimum acceptable DBS diameter of ≥7 mm to ≥8 mm on the AVRR in the individual UK laboratories is shown in Figure 4. All 16 UK laboratories agreed to participate in the exercise, and all 16 laboratories returned results. The mean (range) poor quality AVVR (%) using existing laboratory criteria of ≥7 mm was 2.1% (0.55–5.5). However, when laboratories used ≥8 mm as the minimum acceptable DBS diameter, the mean (range) AVRR increased nearly four-fold to 7.8% (0.55–22.7). Extrapolating this increase in the AVRR observed during the two-week study period to the total number screened during 2020/2021 (689,794) indicates that the implementation of ≥8 mm diameter as the minimum standard for specimen rejection into routine practice would result in an increase of ~39,000 repeat specimen collections and analyses per year until DBS quality had improved. A significant inverse association was observed between the mean number of specimens rejected in the DBS quality assessment distributions 1 and 2 and the predicted AVVR (%) if the ≥8 mm diameter was used as the minimum acceptable criteria; r = −0.734, *p* = 0.003 (Figure 5a).

Following the dissemination of a detailed report on the findings of the initial two EQA distributions in June 2019, and the audit to assess the impact of moving to a minimum DBS diameter size of ≥8 mm, an increase in the number of routine NBS specimens being rejected was reported as part of the yearly U.K. data performance collection [3,16,17]. The breakdown of the number of specimens rejected during the period of 2018/19 to 2020/21 and the reasons for rejection are shown in Figure 6. The number of DBS specimens reported as being compressed/damaged in 2018/2019 was 1449, and this increased to 2798 in 2019/2020. Furthermore, the number of specimens identified as having blood incorrectly applied to the filter paper collection device increased from 2440 in 2018/2019 to 4598 in 2019/2020.

### 3.3. Assessment of the Impact of the National Standard Operating Procedure (SOP) and Using ≥7 mm as the Minimum Acceptable Diameter on the AVRR in the U.K. NBS Laboratories

The impact of the new SOP on the AVRR in the individual U.K. laboratories is shown in Figure 7. All 16 U.K. laboratories returned results. The mean (range) poor quality AVVR (%) using existing laboratory criteria was 2.3% (0.63–5.3). However, when laboratories used the new SOP, the mean (range) AVRR increased nearly three-fold to 6.5% (1.0–20.9).

A significant inverse association was observed between the mean number of specimens rejected in the DBS quality assessment distributions 1 and 2 and the predicted AVVR (%) if the new DBS quality guidelines and SOP were introduced; r = −0.651, *p* = 0.01 (Figure 5b).

## 4. Discussion

Good-quality DBS specimens are vital to ensure that babies with rare but serious conditions are identified and treated early. Good-quality specimens should be obtained the first time to prevent the need for avoidable repeats, as they can cause anxiety for parents, distress to babies, delays in the screening process and lead to both false positive and false negative cases [2,18]. In addition, they are also a waste of scarce healthcare resources.

The initial EQA exercise demonstrated that laboratories inconsistently apply the current 2015 rejection criteria, indicating that the existing standards/guidelines were not being interpreted in the same manner. This finding also raises concerns over the validity of using the AVRR to compare DBS specimen quality performance across the U.K. In fact, our results suggest that counter-intuitively, a low AVVR was indicative of poor-quality specimens being received into laboratories.

The guidelines implemented in 2015 stipulated a minimum spot diameter of ≥7 mm to enable two 3.2 mm sub-punches to be taken from a single DBS specimen. However, the acceptable number of DBS which meet these criteria on an individual NBS card was not stipulated and inevitably led to variation in the rejection rates. The guidelines also stipulated that compressed and multi-spotted specimens should be rejected, but that multi-layered specimens can be accepted provided that they are not spotted onto both sides of the card. Our study demonstrated that many laboratories were unable to identify compressed DBS specimens, differentiate a multi-spotted versus a layered specimen, whether or not the blood had been applied to both sides of the filter paper collection device and specimens where a sufficient amount of blood had not been applied to fully permeate the filter paper, but large enough to allow two sub-punches to be taken.

A major limitation of the 2015 guidelines was that a visual guide of poor-quality specimens was not provided. The inconsistency in applying the DBS quality acceptance guidelines across the laboratories could therefore be attributed to a lack of awareness of the appearance of a poor-quality specimens, e.g., multi-spotted versus a layered specimen or a compressed specimen. This lack of understanding of what constitutes a poor-quality specimen is further supported by data from the annual U.K. data performance reports [3,16,17], which showed an increase in the number of specimens being rejected in the year following the distribution of the DBS EQA images for assessment and the subsequent report. The number of DBS specimens reported as being compressed/damaged in the year preceding the initial study in 2018/2019 was 1449, and this increased to 2798 in the year of the study in 2019/2020. However, it should be noted that more than 5000 babies require a repeat DBS specimen to be collected every year due to missing or incorrect information being recorded on the card.

It should be recognized that screening analytes can be up to 45% lower in compressed DBS specimens and result in the greatest risk of a disorder being missed if concentrations in affected individuals are near to the screening action values [2]. Visually, compressed DBS specimens contain a pale center with a darker ring around the periphery (Figure 2, DBS image 8) which can be difficult to detect in practice [2,6]. However, there are often clues to indicate that the sample is compressed, such as the presence of blood on the glassine envelope and evidence of finger/glove print marks in blood around the DBS. Although the Clinical and Laboratory Standards Institute (CLSI) standards for DBS specimen collection provide a visual listing and description of good- and poor-quality specimens [8], the visual list is not exhaustive and does not encompass other problem specimen collection issues based upon local or country policies.

In response to findings from the initial EQA assessment study, a U.K. group was convened to establish evidence-based standards for minimum DBS specimen diameter and quality to ensure that reliable results are produced. It was agreed by the group that before any changes were implemented, it was essential to achieve the following: (1) improve the consistency with which laboratories assess quality and demonstrate this improvement and (2) improve the existing messaging to specimen takers about the importance of specimen quality and have a clear training strategy agreed to coincide with the launch of the new stricter criteria.

A key priority for the group was to develop an SOP that contained images of good and poor-quality DBS specimens in order to enable a standardized approach of visual assessment and to establish standardized puncher settings to guide minimum DBS size rejection. Several studies have now shown that DBS specimens <8 mm in diameter are associated with a significant negative bias for screening analytes, and this supported a change to a minimum size for the acceptance of ≥8 mm to ensure reliable screening results [2,4]. Studies have also demonstrated that DBS >14 mm (75 µL of blood) produced significant positive biases for screening analytes, which could lead to an increased number of false-positives cases, especially where absolute values of analytes are used in algorithms and where the results in affected infants may be near to the screening cut-offs (e.g., TSH for CHT, Leucine for MSUD, and C5DC for GA1). Furthermore, there is evidence documenting that false positive cases are associated with increased parental anxiety and stress, with increased hospital visits for the infant even after follow-up diagnostic tests have excluded a disorder [19,20]. DBS specimens >14 mm indicate that the specimen has been formed from more than a single drop of blood (Appendix A) and is therefore deemed poor practice, of which we do not want to perpetuate or endorse. It could also indicate that the sample taker is using a non-approved lancet, i.e., not suitable for a baby’s heel, and this may cause damage to the baby’s heel. It should also be noted that the CLSI standards describe such sized samples as being over filled and unacceptable. It was, therefore, recommended that DBS >14 mm should also be rejected.

The results from the study assessing the impact of moving from a minimum acceptable DBS diameter of ≥7 mm to ≥8 mm indicated that the AVRR would increase to an unacceptable level. In particular, there were three laboratories where the AVRR would increase above 15%. Extrapolating data from this study demonstrated that ~39,000 additional repeat specimens would be required every year until the DBS collection technique had improved. Concerns were raised in response to these findings and the fact that many laboratories reported that the majority of specimens being received were barely meeting the existing local criteria. This made the group extremely cautious about implementing more stringent rejection criteria during the winter period, with an increased number of cases of COVID being reported at the time, considering the impact this would then have on the U.K. maternity system. In addition, concerns were also raised that there is a lack of consistency in the rejection of DBS specimens within individual laboratories. Therefore, instead of moving to a minimum acceptable diameter of ≥8 mm, it was decided to modify the SOP to include the existing minimum acceptable diameter of ≥7 mm and to assess the impact of the visual guide SOP to improve the consistency with which the laboratories assess quality. The resulting information would then be used to identify any potential ‘problem’ areas/regions to assist the Clinical Regional Commissioning and Quality Assurance teams to focus resources on improving specimen collection and to help plan the future introduction of the more stringent standards.

Results from the retrospective audit clearly indicate that implementation of the SOP, based upon the 2015 guidelines and using the minimum acceptable diameter of ≥7 mm, would lead to an unacceptable increase in the AVRR in six laboratories. Those laboratories where the AVRR significantly increased (≥8%) were asked to provide additional information and images of specimens that were deemed to be the cause of the observed increase in the AVRR. The following poor-quality specimens resulted in the increased rejection rate:Multi-spotting of blood to fill the circle on the filter paper.Poor application of blood leading to insufficient blood to fully saturate filter paper but large enough to allow two 3.2 mm sub-punches to be taken (Appendix A).Contaminated/water damaged specimens.Ridged/wrinkled specimens.

These images were then independently assessed by the laboratory experts on the DBS quality group, and it was agreed that there is a marked inconsistency in applying the 2015 blood spot quality guidelines and that significant changes would be required to improve DBS quality in those regions.

The evidence for rejecting multi-spotted specimens, and specimens formed from the poor application of blood where there is insufficient blood to fully saturate the filter paper, is based on the fact that analyte results in such specimens are heterogenous due to the non-uniformity of blood distribution across and through the filter paper [2]. In terms of contaminated/liquid damaged DBS specimens, many laboratories would accept specimens if the DBS that was being punched did not show any evidence of contamination/liquid damage. However, it has been demonstrated that specimens subjected to moisture/humidity result in the degradation of analytes within 24 h [13]. In view of this evidence, a decision was made to reject all specimens which include any DBS that show evidence of contamination and or liquid damage. The existing 2015 guidelines (Appendix A) state that ridged/crinkled specimens should be rejected, as this can compromise NBS results [8] (see Appendix A for examples of crinkled specimens). The only way to consistently create ridged/crinkled specimens is to place the filter paper collection device into the glassine envelope within 30 min of blood application, indicating that the specimens have not been appropriately dried. It should be noted that the drying time will be influenced by temperature and humidity. Therefore, it was agreed that the SOP should stipulate that any ridged/crinkled specimen or a specimen with evidence of moisture/liquid exposure or contamination on a DBS or any part of the filter paper should be rejected. Further work is required to understand the impact of various types of poor-quality specimens, in particular the effect of the crinkled/ridged specimens on screening analyte concentrations.

Assessment of DBS specimen quality is highly subjective. Although guidelines were used to aid DBS quality assessment, each laboratory assessed their own specimens and may have interpreted guidelines differently. Rejection rates may have differed if the specimens had been assessed by a single observer or system. In addition, the number of sub-punches obtainable from each DBS was used to estimate DBS size; however, the relationship between DBS size and number of sub-punches can vary between punching instruments. A recent study has shown that the use of a computer vision algorithm is able to accurately measure DBS diameter and identify incorrectly applied blood to the filter paper using images from the Revvity Panthera puncher [21]. Such an approach could be used to complement the existing process for DBS rejection and to address the inconsistency in applying rejection criteria to improve harmonization both within laboratories and between laboratories. Furthermore, this may also be a good way to monitor the success of DBS quality improvement initiatives by providing an objective measure of DBS quality over time and between different laboratories.

The findings from this study have implications for the expansion of the NBS program in the U.K. Increasing the number of disorders that cannot be multiplexed as part of IMD screening would require additional sub-punches for initial screening. The addition of a number of disorders will result in more screen positive results requiring duplicate sub-punches to confirm the initial screening results. Furthermore, good-quality sub-punches are also required to undertake accurate second-tier testing that is required to reduce the false positive rate to maintain an acceptable positive predictive value for the screening test. At present, several laboratories can only just about obtain 4 sub-punches for many of the specimens routinely received and would be unable to perform additional analyses to complete the appropriate screening pathway and confirm the result.

## 5. Conclusions

As part of this quality improvement project, it is recognized that obtaining good-quality DBS specimens is a demanding and difficult task that requires considerable skill and expertise. The implementation of regular training programs, providing educational resources for specimen takers and ensuring that the importance of DBS specimen size and quality is understood, is essential to improving and maintaining the quality of DBS specimen collection. Furthermore, an evidence-based ‘national’ SOP with clear visual examples of specimens that should be rejected is required to maintain a consistent set of standards when assessing the suitability of DBS specimens for analysis and to ensure consistency in specimen acceptance/rejection both within and between laboratories. Finally, the development of a DBS quality EQA scheme is essential to assess the consistency of specimen acceptance/rejection in newborn screening laboratories.

## Figures and Tables

**Figure 1 IJNS-10-00060-f001:**
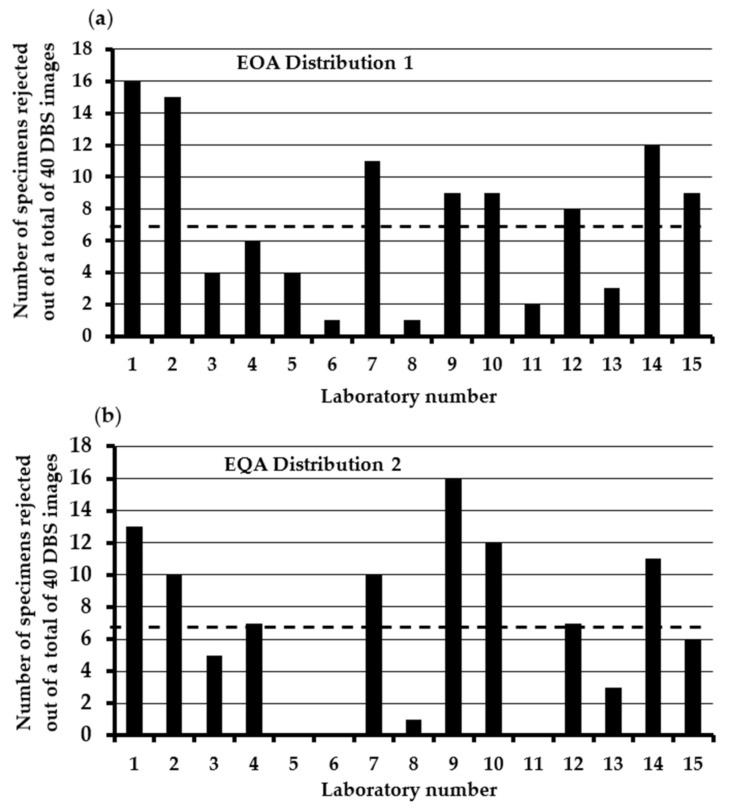
The number of DBS specimens rejected by each laboratory in (**a**) EQA distribution 1 and (**b**) EQA distribution 2. Dotted lines show the mean number of specimens rejected. A total of 40 DBS images were disseminated for each distribution. The laboratory numbers are the same in both distributions. NB—Laboratory 11 did not return results in EQA distribution 2.

**Figure 2 IJNS-10-00060-f002:**
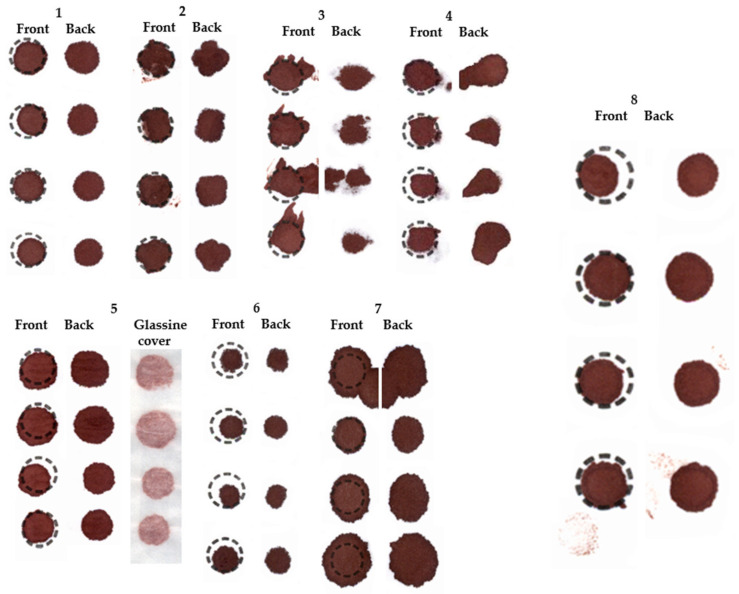
Examples of DBS specimen images included in the two separate EQA distributions. Image 1 shows a good-quality specimen (0/15 labs rejected this specimen), image 2—multi-spotted (11/15 labs rejected), image 3—poor quality/uneven saturation of blood (8/14 labs rejected), image 4—blood applied to both sides of filter paper (9/14 labs rejected), image 5—wet specimen placed in glassine envelope/compressed (10/14 labs rejected), image 6—Insufficient/too small (9/14 labs rejected), image 7—excess blood/layering (2/15 labs rejected), image 8; a compressed specimen (6/15 labs rejected).

**Figure 3 IJNS-10-00060-f003:**
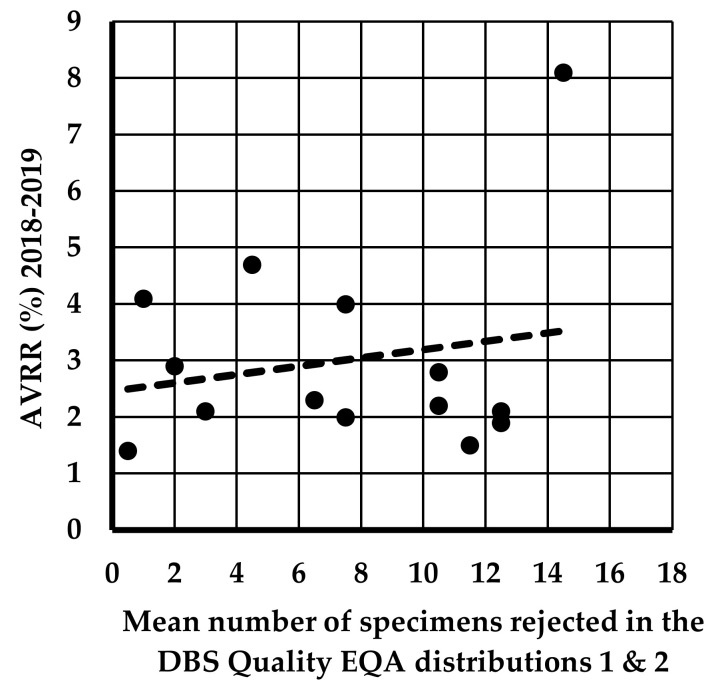
The relationship between the mean number of specimens rejected in the DBS quality assessment distributions 1 and 2 and the UK AVVR (%) in the U.K. during the period of the study (2018–2019); r = 0.194, *p* = 0.506.

**Figure 4 IJNS-10-00060-f004:**
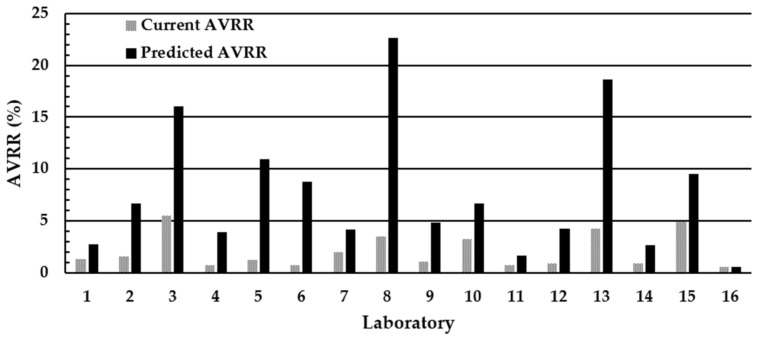
Impact of implementing the ≥8 mm minimum DBS specimen diameter rejection criteria on the AVRR in the 16 U.K. NBS Laboratories. The AVRR is calculated from DBS specimens routinely received into the individual laboratories. The laboratory numbers are the same in all assessments.

**Figure 5 IJNS-10-00060-f005:**
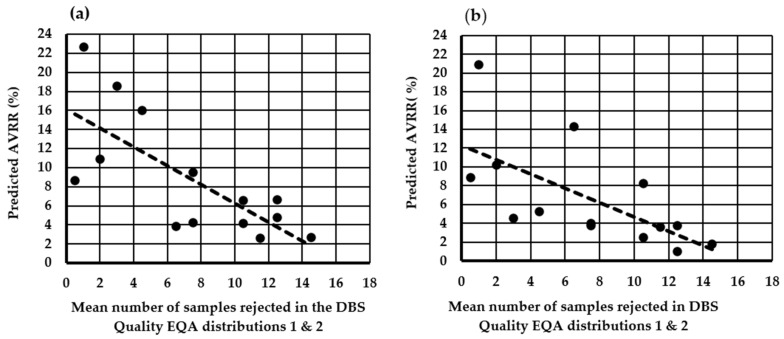
The relationship between the mean number of specimens rejected in the DBS quality EQA assessment distributions 1 and 2 with (**a**) the predicted AVVR (%) if the minimum diameter of ≥8 mm was introduced; r = −0.734, *p* = 0.003 and (**b**) the predicted AVVR (%) if the new visual guide SOP was introduced and using the existing ≥7 mm as the minimum acceptable standard; r = −0.651, *p* = 0.01.

**Figure 6 IJNS-10-00060-f006:**
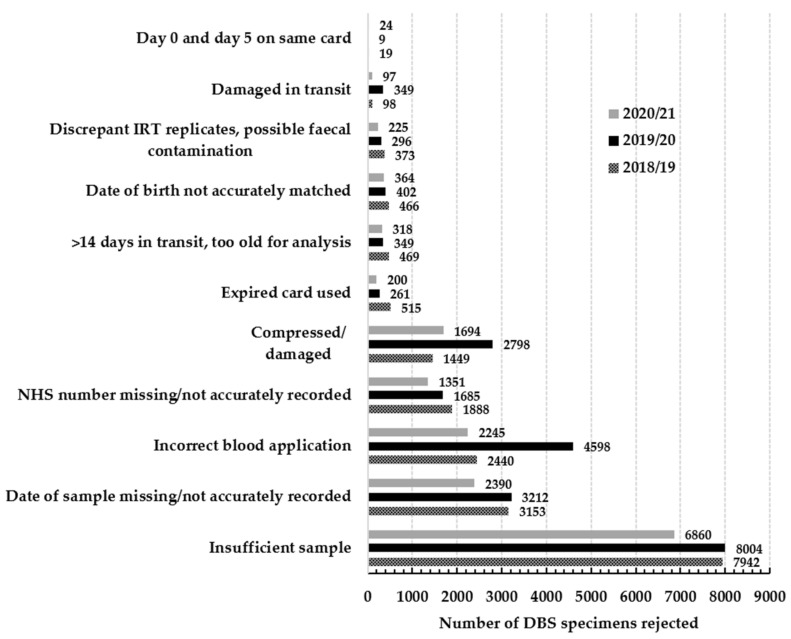
Breakdown of DBS specimens rejected in the U.K. during the period 2018–2021. The number of babies screened during 2018/2019, 2019/2020 and 2020/2021 was 741,577, 723,295 and 689,794 respectively. NB—these figures are based upon the 2015 rejection criteria [Appendix A].

**Figure 7 IJNS-10-00060-f007:**
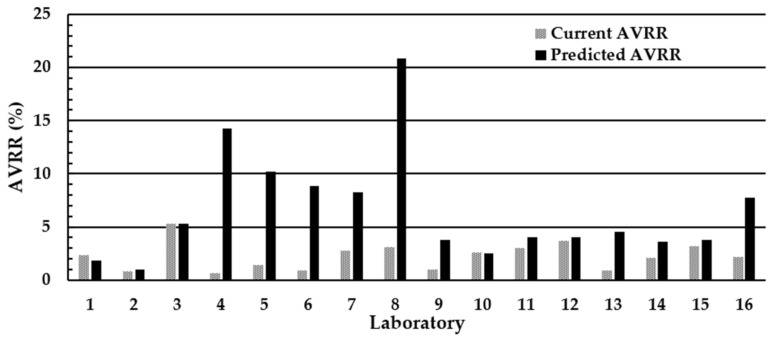
Impact of implementing the visual guide SOP based upon the 2015 criteria and using ≥7 mm as the minimum acceptable diameter on the AVRR in the U.K. NBS laboratories. The laboratory numbers are the same in all assessments.

## Data Availability

The data presented in this study are available on request from the corresponding author.

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
