# Peer review of "Consistency in the Assessment of Dried Blood Spot Specimen Size and Quality in U.K. Newborn Screening Laboratories"

_2409-515X, 2024, doi:10.3390/ijns10030060_

Round 1
Reviewer 1 Report
Comments and Suggestions for Authors
The report of the UK survey of DBS rejection is very important for the quality of NBS.
Minor comments
Page 5. For better understanding of Figure 1, a mention should be added concerning the number of BDS specimens rejected from a set of 40 images of BDS for each distribution.
Page 6. Figure 2 : first line of legend text is not completly readable.
Pages 6-7. Figure 3 and its legend should be on a same page.
Page 7. Figure 7 legend should mention that the AVRR is calculated from BDS routinely received into each lab.
Author Response
Reviewer 1
Minor comments
Page 5. For better understanding of Figure 1, a mention should be added concerning the number of BDS specimens rejected from a set of 40 images of BDS for each distribution.
We have amended Figure 1 legend on page 5, lines 218-219 to include the following “A total of 40 DBS images were disseminated for each distribution”. In addition, we have amended the axis labels on the graphs to include the following “Number of specimens rejected out of a total of 40 DBS images”
Page 6. Figure 2 : first line of legend text is not completely readable.
Thank you for highlighting this formatting issue. We have amended the spacing on Page 6 to correct this.
Pages 6-7. Figure 3 and its legend should be on a same page.
Thank you for highlighting this formatting issue. We have amended the spacing on Page 6 to correct this.
Page 7. Figure 7 legend should mention that the AVRR is calculated from BDS routinely received into each lab.
We assume that the reviewer was referring to Figure 4 legend?!
Thank you for pointing this out. We have amended the figure legend on page 7, lines 235-236 to include “The AVRR is calculated from DBS specimens routinely received into the individual laboratories. “
Reviewer 2 Report
Comments and Suggestions for Authors
This well-written manuscript provides a detailed description of the NBS DBS quality assessment approaches and improvement processes in the UK newborn screening laboratories. The manuscript is well structured, making the content easy to follow. Sharing this UK experience will be highly beneficial also for other NBS programs, especially those with multiple screening centers. In addition, the SOP with the color images of the poor-quality DBS specimens (Supplementary Section) will be a great educational resource for NBS labs looking to improve the DBS quality assessment in their centers.
General concept comment
The authors may consider adding a brief introductory paragraph describing the UK NBS program in the Introduction section (i.e., number of screening labs, conditions screened for, second tier testing (if any), and number of infants screened per year). Although most of this information is described separately in different sections of the manuscript, having it summarized in one place will make it easier for readers to understand the “Materials and Methods” and “Results” sections.
Specific (minor) comments:
Comment #1: L78; Page 2 (Introduction)
Please explain the meaning of 2018/2019, as this may not be clear to all readers (whether it represents a single fiscal year or a span of two years).
Comment #2: L138 -140; Page 3 (Materials and Methods 2.2.1)
“It was agreed that the minimum and maximum acceptable DBS diameter would be ≥8 and ≤14mm, respectively to allow a minimum of six sub-punches to be taken.”
This is not completely clear; a minimum of six sub-punches from a single spot? (Later in this section the authors mention that only three 3.2 mm sub-punches can be obtained from an 8mm diameter DBS (using the standardized puncher settings).
Comment#3: Results 3.1 and Figure 1
In both EQA distributions, 15 out of the 16 UK laboratories agreed to participate. My understanding is that these 15 labs are the same in both distributions. Can this be confirmed?
In the EQA distribution 2, only 14/15 labs returned results (Lab-11 did not return the results). Figure 1b: Please remove the non-participant lab (Lab-11) from the x-axis.
Comment#4: L203-205; Page 5 (Results 3.2)
“The implementation of the ≥8mm minimum diameter for specimen rejection into routine practice would result in an increase of ~41,000 repeat specimen collections and analyses.”
What percentage of the total specimens would this increase represent? What is the time interval for this increase?
Comment#5: Figure 2; Page 6
Please note that in the PDF copy I downloaded, Fig 2 partially obscures the legend for this figure.
Comment#6: Figure 5 Legend; Page 8
To provide better clarity, please include the rejection criteria used in 2018 -2021 in the description.
Comment #8 Please review the order of your figures. Figures should be numbered in the order they are first mentioned in the text; Figure 7a (Page 5) comes before Figures 5 and 6 (both Page 7).
Author Response
Reviewer 2
This well-written manuscript provides a detailed description of the NBS DBS quality assessment approaches and improvement processes in the UK newborn screening laboratories. The manuscript is well structured, making the content easy to follow. Sharing this UK experience will be highly beneficial also for other NBS programs, especially those with multiple screening centers. In addition, the SOP with the color images of the poor-quality DBS specimens (Supplementary Section) will be a great educational resource for NBS labs looking to improve the DBS quality assessment in their centers.
General concept comment
The authors may consider adding a brief introductory paragraph describing the UK NBS program in the Introduction section (i.e., number of screening labs, conditions screened for, second tier testing (if any), and number of infants screened per year). Although most of this information is described separately in different sections of the manuscript, having it summarized in one place will make it easier for readers to understand the “Materials and Methods” and “Results” sections.
Thank you for highlighting this - We have now included a paragraph in the Introduction on page 2 (lines 70-72) which outlines the U.K. Newborn screening programme and helps sets the scene. “In the U.K. there are 16 laboratories (13 in England, 1 in Wales, 1 in Scotland and 1 in Northern Ireland) with ~700K babies screened per year for 9 conditions.“
Specific (minor) comments:
Comment #1: L78; Page 2 (Introduction)
Please explain the meaning of 2018/2019, as this may not be clear to all readers (whether it represents a single fiscal year or a span of two years).
We have now inserted (fiscal year) after 2018/2019 on line 80 on page 2.
Comment #2: L138 -140; Page 3 (Materials and Methods 2.2.1)
“It was agreed that the minimum and maximum acceptable DBS diameter would be ≥8 and ≤14mm, respectively to allow a minimum of six sub-punches to be taken.”
This is not completely clear; a minimum of six sub-punches from a single spot? (Later in this section the authors mention that only three 3.2 mm sub-punches can be obtained from an 8mm diameter DBS (using the standardized puncher settings).
Thank you for pointing this out - We have now re-phrased the section on lines 141-144.
“It was agreed that the minimum and maximum acceptable DBS diameter would be ≥8 and ≤14mm, respectively. A minimum DBS diameter of ≥8mm would enable a minimum of six sub-punches to be taken from two DBS.”
Comment#3: Results 3.1 and Figure 1
In both EQA distributions, 15 out of the 16 UK laboratories agreed to participate. My understanding is that these 15 labs are the same in both distributions. Can this be confirmed?
In the EQA distribution 2, only 14/15 labs returned results (Lab-11 did not return the results). Figure 1b: Please remove the non-participant lab (Lab-11) from the x-axis.
Thank you for highlighting this - To clarify the point regarding the laboratory numbers we have inserted the following in Figure 1 legend “The laboratory numbers are the same in both distributions”.
Furthermore, I have also added the following to the legends of Figures 4 & 7 – “The laboratory numbers are the same in all assessments.”
By removing the lab 11 from the axis of figure 1b it shifts all the results and it makes it very difficult for the reader to compare laboratory performance between distribution 1 and 2. We have therefore not removed it from figure 1b. Please note that the figure legend clearly states that laboratory 11 did not return results in EQA distribution 2.
Comment#4: L203-205; Page 5 (Results 3.2)
“The implementation of the ≥8mm minimum diameter for specimen rejection into routine practice would result in an increase of ~41,000 repeat specimen collections and analyses.”
What percentage of the total specimens would this increase represent? What is the time interval for this increase?
Thank you for highlighting this – This figure of ~41,000 was an extrapolated figure based upon the number of DBS specimens routinely received during a two week period in November 2022. In hindsight this was not the correct thing to do. We have now recalculated based upon the true number screened during the period 2020/21 (689,794)
We have now inserted the following on lines 206-211 on page 5 to clarify.
“Extrapolating this increase in the AVRR observed during the two week study period to the total number screened during 2020/2021 (689,794), indicates that the implementation of ≥8mm diameter as the minimum standard for specimen rejection into routine practice would result in an increase of ~39,000 repeat specimen collections and analyses per year until DBS quality had improved.”
Comment#5: Figure 2; Page 6
Please note that in the PDF copy I downloaded, Fig 2 partially obscures the legend for this figure.
We have amended the spacing on Page 6 to correct this.
Comment#6: Figure 5 Legend; Page 8
To provide better clarity, please include the rejection criteria used in 2018 -2021 in the description.
We have now inserted the following into the Figure legend on page 8 - “NB – these figures are based upon the 2015 rejection criteria [Figure S1].”
Please note that this is now figure 6 in manuscript.
Comment #8 Please review the order of your figures. Figures should be numbered in the order they are first mentioned in the text; Figure 7a (Page 5) comes before Figures 5 and 6 (both Page 7).
Thank you for pointing this out. We have now re-ordered the figures in the manuscript.